# Advanced Cell Culture Models Illuminate the Interplay between Mammary Tumor Cells and Activated Fibroblasts

**DOI:** 10.3390/cancers15092498

**Published:** 2023-04-26

**Authors:** Martina Del Nero, Alessandro Colombo, Stefania Garbujo, Chiara Baioni, Linda Barbieri, Metello Innocenti, Davide Prosperi, Miriam Colombo, Luisa Fiandra

**Affiliations:** Department of Biotechnology and Biosciences, University of Milano-Bicocca, Piazza della Scienza 2, 20126 Milan, Italy

**Keywords:** advanced cell culture models, triple-negative breast cancer, CAF-cancer cell crosstalk, TGF-β, PDGF, IL-6, transwell, spheroids

## Abstract

**Simple Summary:**

Desmoplastic tumors are known to be highly aggressive and difficult to eradicate. In these tumors, the interactions between tumor cells and their microenvironment are pivotal in regulating cancer progression. In particular, activated fibroblasts and/or cancer-associated fibroblasts (CAFs) play a key role in cancer-cell invasiveness and in resistance to chemotherapy. Here, we validated the usage of advanced cell culture models as tools to study the interplay between mammary tumor cells and fibroblasts. As a proof of principle, we investigated the mechanism of action of soluble factors involved in breast cancer progression, such as TGF-β, PDGF, and IL-6. We posit that advanced transwell and spheroid co-cultures provide a pathologically relevant tractable system to study the role of the tissue micro-environment in cancer progression.

**Abstract:**

The interaction between tumor cells and activated fibroblasts determines malignant features of desmoplastic carcinomas such as rapid growth, progression towards a metastatic phenotype, and resistance to chemotherapy. On one hand, tumor cells can activate normal fibroblasts and even reprogram them into CAFs through complex mechanisms that also involve soluble factors. Among them, transforming growth factor beta (TGF-β) and Platelet-Derived Growth Factor (PDGF) have an established role in the acquisition of pro-tumorigenic phenotypes by fibroblasts. On the other hand, activated fibroblasts release Interleukin-6 (IL-6), which increases tumor-cell invasiveness and chemoresistance. However, the interplay between breast cancer cells and fibroblasts, as well as the modes of action of TGF-β, PDGF, and IL-6, are difficult to investigate in vivo. Here, we validated the usage of advanced cell culture models as tools to study the interplay between mammary tumor cells and fibroblasts, taking mouse and human triple-negative tumor cells and fibroblasts as a case study. We employed two different settings, one permitting only paracrine signaling, the other both paracrine and cell-contact-based signaling. These co-culture systems allowed us to unmask how TGF-β, PDGF and IL-6 mediate the interplay between mammary tumor cells and fibroblasts. We found that the fibroblasts underwent activation induced by the TGF-β and the PDGF produced by the tumor cells, which increased their proliferation and IL-6 secretion. The IL-6 secreted by activated fibroblasts enhanced tumor-cell proliferation and chemoresistance. These results show that these breast cancer avatars possess an unexpected high level of complexity, which resembles that observed in vivo. As such, advanced co-cultures provide a pathologically relevant tractable system to study the role of the TME in breast cancer progression with a reductionist approach.

## 1. Introduction

Breast carcinomas are a complex cellular community consisting of different cell types that interact with each other. The tumor microenvironment (TME) determines both cancer progression and the response to treatment. Cancer-Associated Fibroblasts (CAFs) are particularly abundant in the TME of desmoplastic tumors (i.e., pancreatic adenocarcinoma, bladder cancer, small-cell lung cancer, and some breast cancer subtypes) and contribute to making desmoplastic tumors highly aggressive and hard to eradicate. CAFs affect the remodeling of the extracellular matrix (ECM)—the so-called desmoplastic reaction [1]—and Epithelial-to-Mesenchymal Transition (EMT) in tumor cells [2,3], which may increase their proliferation, migration and invasion, and chemoresistance [4,5]. Altogether, these effects stimulate cancer progression. CAF activation is the result of cancer cells recruiting resident normal fibroblasts and converting them into to tumor-sustaining cells [6].

Interplay between cancer cells and CAFs involves ECM components and integrins [7], as well as paracrine signals such as exosomes and soluble mediators (e.g., growth factors and cytokines), some of which have recently been chosen as targets for the development of new anti-cancer therapeutics [8]. Among the growth factors, transforming growth factor beta (TGF-β) and Platelet-Derived Growth Factor (PDGF) play an important role in the reprogramming of normal fibroblasts into CAFs [9]. The interaction between tumor cells and resident fibroblasts often hyperactivates TGF-β1 signaling in the latter cells, which become α-SMA-positive CAFs [10]. SMAD proteins are phosphorylated upon activation of TGF-β receptor (TGF-βR) by its specific ligand TGF-β and phosphorylated SMAD proteins subsequently translocate into the nucleus and function as transcription factors inducing the expression of CAF-related genes, including αSMA [11]. PDGF released by cancer cells has been shown to stimulate the proliferation of CAFs, promoting a desmoplastic reaction in tumors [12]. PDGF acts on stromal fibroblasts by binding to and activating tumor-specific PDGFR signaling cascades [13,14].

Activated CAFs enhance tumor cells’ invasiveness and resistance to chemotherapy through the release of soluble mediators such as growth factors [9], among which TGF-β1 has a main role in PDAC [15]. Other factors promoting cancer progression and chemoresistance are chemokines (i.e., SDF-1/CXCL12) [16,17] and cytokines, such as interleukine-6 (IL-6). IL-6 expression and release are increased in fibroblasts associated with gastric cancers compared to normal fibroblasts [18]. The IL-6 secreted by CAFs enhances migration, invasion, EMT and chemoresistance in lung [19,20], gastric [18] and prostate [21] cancer cells. IL-6-induced EMT and chemoresistance have also been demonstrated in luminal breast cancers [22] via a mechanism that involves the IL-6/JAK/STAT3 signaling axis, one of the most promising therapeutic targets for cancer control under clinical investigation [23].

Triple-negative breast cancer (TNBC) remains hard to eradicate, due to the lack of effective targeted therapies and resistance to standard chemotherapeutic regimens, and it is associated with a poor prognosis. The high metastatic capacity of TNBC is inherent in the desmoplastic nature to which CAFs lend a key contribution. It is the cross-talk between CAFs and cancer cells that plays a pivotal role in the aggressive profile of this breast cancer type, and different prognostic factors or potential therapeutic targets implicated in this crosstalk have been identified [24,25]. Therefore, therapeutic strategies based on CAF targeting have been proposed as an appealing approach to breast cancer (BC) management. Concerning the factors secreted by CAFs and involved in TNBC growth and progression, hepatocyte growth factor [26] and SDF-1/CXCL12 [27] may have a role in other BC subtypes. Stromal cytokines, IL-6 in particular, are known to activate tumor cells [23,28], but it is hitherto unclear whether and how increased TNBC aggressiveness and chemoresistance are specifically induced by CAFs.

Here, we set out to investigate this issue by taking advantage of advanced cell culture models (described in Appendix A) and illuminating the interplay between TNBC tumor cells and associated fibroblasts, dissecting the contribution of TGF-β, PDGF and IL-6. We used murine mammary triple-negative tumor cells (4T1 cells) and murine fibroblasts (NIH-3T3 cells) as a test case, and employed two different settings, one permitting only paracrine signaling (transwell), the other both paracrine and cell-contact-based signaling (spheroid).

We found that the TGF-β and PDGF secreted by the tumor cells activated the fibroblasts, which increased their proliferation and IL-6 secretion. In turn, the IL-6 secreted by activated fibroblasts enhanced tumor-cell proliferation and chemoresistance. In summary, our results showed that these breast cancer avatars possess a high level of complexity, resembling that observed in vivo. Thus, these advanced co-cultures qualify as pathologically relevant tractable systems to study the interplay between the TME and mammary tumor cells with a reductionist approach.

## 2. Materials and Methods

### 2.1. Cell Culture

Murine triple-negative breast cancer 4T1 cell line (Murine Bioware-Ultra 4T1-Luc2) was originally obtained from Perkin Elmer (Milano, Italy), while murine embryonic fibroblasts NIH-3T3, human lung fibroblasts MRC-5 and human triple-negative MDA-MB-231 cell lines were obtained from the American Type Culture Collection (by LGC Standards, Sesto San Giovanni, Italy). 4T1 cells were cultured in RPMI 1640 medium, supplemented with 1% penicillin/streptomycin (P/S), 1% L-glutamine and 10% fetal bovine serum (FBS). NIH-3T3 cells were cultured in Dulbecco’s Modified Eagle Medium (DMEM) High Glucose medium supplemented with 1% penicillin/streptomycin (P/S), 1% L-glutamine and 10% calf bovine serum (CBS). MRC-5 and MDA-MB-231 cells were cultured in the same medium with 10% fetal bovine serum (FBS), instead of CBS. All cell lines were grown at 37 °C, 5% CO_2_ and controlled humidity. Cell culture medium and chemicals were purchased from EuroClone (Pero, Italy). Cell growth and health was monitored daily by Paula digital microscope (Leica Microsystems, Buccinasco, Italy).

### 2.2. Reagents for Cell Treatment

For murine embryonic fibroblasts NIH-3T3 cell line, transforming growth factor TGF-β (ab50036, Abcam, Cambridge, UK) and platelet-derived growth factor PDGF-BB (SRP3229-10 μg, Merck, Milano, MI, Italy) were used. Interleukin-6 (RMIL6I, Invitrogen, Waltham, MA, USA) was applied to the murine breast cancer 4T1 cell line. The growth factors were maintained at −20 °C in PBS-BSA 2 mg/mL and PBS-BSA 0.1%, for TGF-β and PDGF-BB, respectively. Interleukin IL-6 was maintained at −80 °C in PBS-BSA 1%. The ALK5-selective TGF-βRI receptor inhibitor SB431542 (S1067-10 mg, Selleck, Planegg, Germany), the SMAD3 inhibitor, SIS3 (A16065-5 mg, Adooq Bioscience, Irvine, CA, USA) and the PDGF α/β receptor inhibitor, CP-673451 (HY-12050, MedChemExpress, Monmouth, Junction, NJ, USA) were used for both the cell lines. The inhibitors were kept at −80 °C, and only SB4315432 and SIS3 have been diluted in sterile 100% Ethanol solvent. Doxorubicin hydrochloride (DOX) was purchased by Sigma-Aldrich (Milan, Italy) and diluted in water.

### 2.3. Transwell Mono- and Co-Cultures

Murine 4T1 cells were harvested after trypsinization and seeded in a 12 well plate (15,000 cells/well) in 1.5 mL RPMI medium. Murine fibroblasts NIH-3T3 cells were harvested after trypsinization and seeded on the membrane of 12-Well Cell Culture Inserts, 0.4 µm pore size (ThinCert^®^—Greiner Bio-One, Cassina de’ Pecchi, Italy) (15,000 cells/insert). For the co-culture condition, after one day of monoculture, the inserts bearing the 3T3 were moved into the multiwell with the tumor cells. Control monocultures were performed inserting 3T3 cells on transwell into the well with RPMI medium but in the absence of 4T1, or culturing 4T1 cells without 3T3-bearing inserts. From this moment, growth factors, their inhibitors, or IL-6 were added to the media according to the different experimental conditions. After two days, the media of upper and lower chambers were changed, maintaining their own culture medium and replacing the additive molecules. Co-culture and monocultures in transwells/wells were so kept for further 5 days in incubation (time limit that can be reached to avoid cell sufferance into the transwell/well systems) before proceeding with the following analyses.

### 2.4. Monospheroids and Heterospheroids, and Paula Imaging

For the production of heterospheroids, harvested 4T1 and NIH-3T3 cells were seeded together into each well of ultra-low attachment nucleosphere plate with hemispherical bottom (Nunclon™ Sphera™ 96-Well, Thermo Fisher Scientific, Monza, Italy) at the density of 3000 cells/well for each cell type (6000 total cells) in 100 μL or 200 μL (for IL-6 Elisa detection in supernatants) of a mixed medium RPMI/DMEM (1:1). The plate was centrifuged at 1300 rpm for 10 min and the produced spheroids were observed by Paula. Control monospheroids were performed using 3T3 or 4T1 cells with their own culture medium (3000 total cells). From this moment, growth factors and their inhibitors were added to the culture medium according to the different experimental conditions. Hetereospheroids and monospheroids were so kept for 7 days in incubation (2 days longer than transwell system, in order to obtain the maximal exposure time without incurring in cell sufferance), before proceeding with the following analyses. To produce the heterospheroids starting from the human TNBC cells MDA-MB-231 and the fibroblasts MRC-5, the same number of cells and the same procedure described above was used, but in this case, cells were diluted in Cultrex^®^ (3-D Culture Matrix Reduced Growth Factor BME, R&D SYSTEM, Minneapolis, MN, USA). The matrix was thawed on ice overnight and added at a final concentration of 2.5% with ice-cold pipette tips to the cell suspension, before seeding into the nucleosphere plate.

All monospheroids and heterospheroids were analyzed by Paula digital microscope to determine their formation and diameter in the absence or presence of treatments.

### 2.5. Cell Proliferation Assays

NIH-3T3 cultured on transwell in the absence or presence of growth factors with or without their inhibitors, or in the presence of 4T1 cells with or without the same inhibitors, were harvested from the upper side of the insert and counted in Burker’s chamber. The same procedure was performed with 4T1 cells cultured for 5 days in a 12-wells multiplate (15,000 cells seeded in each well on the first day) in the absence or presence of IL-6 (50 ng/mL), or in the presence of 3T3 cells.

An evaluation of proliferation of the 4T1 cells co-cultured with the NIH-3T3 into the heterospheroids versus the 4T1 in monospheroid, was performed by measuring the bioluminescent signal produced by these luciferase+ cells (4T1-Luc2), after addition of 40 µM D-Luciferin potassium salt (Merck, Milano, Italy) into the culture medium (100 µL in the well of the nucleosphere plate). The bioluminescence intensity (BLI) was recorded over ten minutes from the addition of luciferin, by EnSight™ Multimode Plate Reader (PerkinElmer, Milano, Italy), and was compared to the BLI of untreated mono- and heterospheroids.

### 2.6. Quantification of IL-6 Released by Activated Fibroblasts

The ELISA test was used for the quantification of the cytokine IL-6 released by the activated fibroblasts, cultured in transwell or spheroid, into the culture medium. At the end of the culture period, in which the medium was not changed, supernatants were collected and centrifuged at 12,000 rpm for 10 min at 4 °C. The supernatants were maintained at −20 °C and diluted in culture medium up to 200 µL to be then analyzed by IL-6 Mouse Uncoated ELISA Kit (Life Technologies, Monza, Italy). The 96-wells plate was read with the EnSight multimode plate reader at wavelengths of 450 nm and 570 nm.

### 2.7. Viability Assays

The effect of 0.1 or 1 μM DOX on 4T1 cells co-cultured with 3T3 cells, or in monoculture with or without IL-6 (50 ng/mL), was evaluated by MTS viability test. Untreated 4T1 cells were used as controls. After 48 h of incubation with DOX, the cells were washed with PBS and then incubated for 3 h at 37  °C with 200 μL of fresh RPMI medium and 40 µL of 3-(4,5-dimetiltiazol-2-il)-5-(3-carbossimetossifenil)-2-(4-solfofenil)-2H-tetrazolio (MTS) stock solution. After incubation, 100 μL of the violet solution from the plate was transferred to a new 96-well multiwell plate, in duplicate for each sample. Afterwards, the 490 nm absorbance results were then measured with an EnSight™ and the cell viability was calculated by normalizing the absorbance of the treated samples against the level recorded in the untreated sample.

### 2.8. Analysis of Fluorescence in the Spheroids

For the detection of nuclei fluorescence into the mono- and heterospheroids, 4T1 cells were labeled with 10 μM of Hoechst 33342 dye (Cat 62249, Thermo Fisher Scientific, Monza, Italy), before their assembling. Cells were incubated with Hoechst at 37 °C for 10 min in RPMI medium in the dark. After incubation, the cells were centrifuged at 1200 rpm for 5 min. The pellet was washed 3 times in PBS and the cells were counted in Burker’s chamber and then seeded for spheroids formations. After 7 days, the 4T1/3T3 heterospheroids, exposed or not the TGF-βRI receptor inhibitor, and the 4T1 monospheroids, exposed or not to IL-6 (50 ng/mL), were transferred from the nucleosphere to a 96-well multiwell plate with a flat black base (Greiner Bio-One, Cassina de’ Pecchi, Italy Italy) and kept overnight in incubation before imaging by Operetta CLS fluorescence microscope (Operetta CLS High Content Analysis System—PerkinElmer, USA). After fluorescence acquisition (pre-treatment), 1 μM DOX was added to the spheroids and the plate was kept in incubation for 48 h before the subsequent acquisition. Untreated spheroids were used as controls. The fluorescence intensity of 4T1 nuclei in to the spheroids was quantified by Operetta Harmony™ software 4.8.

For the double-labeling of 4T1 cells and NIH-3T3 cells into the heterospheroid, before its assembling, the two cell types were incubated with CellTracker™ Deep Red dye (λex = 630, λem = 660 nm), and Vybrant^®^ CFDA SE (λex = 492, λem = 517 nm) (Thermo Fisher Scientific, Monza, Italy), respectively. After 7 days of co-culture, the heterospheroid was fixed (10 min in paraformaldehyde 4%), washed in PBS 1X, and analyzed by the fluorescence microscope Thunder Imager 3D Live Cells (Leica Microsystems, Buccinasco, Italy).

The viability of 4T1 cells and NIH-3T3 cells into the large 7-days heterospheroid was detected by exposing it to the nuclear dye Hoechst 33342. In detail, heterospheroids were first permeabilized with Triton-X 0.01% for 5 min, and then exposed to 20 μM Hoechst for 2 h, to be then analyzed by Thunder fluorescence microscope. A single inner z-stack was considered to detect the viability of the cells up to the deep core of the spheroid.

### 2.9. Wound Healing and Migration Assays

Wound healing assay was performed to test the migratory activity of 4T1 cells, after exposure in transwell to NIH-3T3 fibroblasts or IL-6 (50 ng/mL) using culture-insert petri dishes (Culture-Insert 3 Well, Ibidi, Gräfelfing Germany). Harvested cells were seeded at the density of 6 × 104/cm2 inside the 3 small chambers of the petri dish containing the silicon insert and cultured for 24 h in RPMI medium. After the removal of the insert, fresh medium was added and the migration of the cells into the cell free space was monitored at 1, 4, 7 and 21 h by Paula imaging system, and the cell-free space was measured over time (0–7 h), starting from the initial distance of 500 μm.

Migration assay was performed with the same 4T1 cell samples reported above, by seeding them on the membrane of a 12-Well Cell Culture Inserts, 8 µm pore size (ThinCert^®^—Greiner Bio-One, Cassina de’ Pecchi, Italy). The cells (2 × 10^5^) were seeded in 400 µL of medium without FBS into the upper chamber and the same medium was also added into the lower chamber of the transwell system. After 24 h incubation at 37 °C, the medium of lower chamber was replaced with the same RPMI added with FBS 10%. After 48 h the medium into the upper chamber was aspired and the non-migrated cells on the upper surface of the transwell membrane were gently scrapped with a cotton swab. The migrated cells were detected by the colorimetric method CytoSelect™ Cell Migration Assay (Creative Bioarray, Shirley, NY, USA), including cell staining, dye extraction and spectrophotometric quantification. In detail, the inserts were moved to wells containing the stain solution and, after 10 min at RT, were washed in water and put into the extraction solution. After other 10 min, 100 µL of samples were quantified with the EnSight™ at the wavelength of 560 nm.

### 2.10. Statistical Analysis

All statistical analyses are conducted by one-way ANOVA with a multiple comparison using the two-stage linear step-up procedure of Benjamini, Krieger and Yekutieli (PraphPad Prism 9 software). Student’s *t*-test is also applied in a one-to-one comparison of Mean values (Appendix A). Values of *p* < 0.05 were considered statistically significant.

## 3. Results and Discussion

### 3.1. Tumor Cells Promote Activation of Normal Fibroblasts through Paracrine Signaling

#### 3.1.1. Hyperproliferation of Activated Fibroblasts

Activation of NIH-3T3 fibroblasts by soluble factors released by the 4T1 cancer cells was initially evaluated monitoring their proliferation at five days of culture in a transwell unit, with or without tumor cells being present at the lower compartment. We found that the number of NIH-3T3 cells co-cultured with the 4T1 cells was significantly higher than that of the corresponding NIH-3T3 monocultures (Figure 1b). A similar effect was obtained treating the NIH-3T3 cells with TGF-β and this was partially reversed by adding SB431542, a TGF-β receptor (TGF-βR) inhibitor (Figure 1a). Interestingly, SB431542 also reduced the stimulatory effects of the factors secreted by the 4T1 cells (Figure 1b). These observations suggest that TGF-β is one of the signals whereby 4T1 cells bolster fibroblast proliferation. Of note, SMAD3 inhibition by SIS3 fully blunted NIH3-T3 hyperproliferation in the co-cultures, consistent with SMAD3 being a convergence point of different signaling pathways that mediate tumor cell-induced activation of fibroblasts [29].

To assess the reach of the above observations, we took advantage of PDGF, another mediator often released by BC cells. A significant increase in the number of NIH-3T3 cells could be observed upon addition of PDGF-BB, even though the effect was less prominent than that of TGF-β (Figure 1a). As expected, this was reversed by the PDGF receptor inhibitor CP-673451. More importantly, CP-673451 also markedly reduced fibroblast number in the co-cultures (Figure 1b), showing that the 4T1 cells can use both PDGF and TGF-β to alter the proliferative abilities of the NIH-3T3 cells.

Increased proliferation is a key feature of activated fibroblasts and CAFs; thus, it was evaluated also using spheroids made of NIH-3T3 fibroblasts either alone (monospheroids) or in combination with 4T1 tumor cells (heterospheroids). By labeling the two cell types with different fluorescent dyes before heterospheroid formation, we could demonstrate their homogenous distribution in the resulting 3D structure and the absence of a hollow core (Appendix A). Moreover, labeling the heterospheroids with the nuclear dye Hoechst suggested that, despite the dimensions (500–800 µm), the innermost cells were viable (Appendix A).

TGF-β allowed monospheroids to reach a significantly increased size, which was fully reversed by the TGF-βR inhibitor (Figure 2). PDGF-BB had a similar stimulatory effect, suggesting that it can modulate NIH-3T3 number also in the spheroid cultures. Overall, this is in line with the results of the transwell experiments. The heterospheroids were considerably larger in size than the monospheroids, because they consist of two cell populations. Nevertheless, both inhibitors reduced the spheroid diameter, the TGF-βR inhibitor being more effective (Figure 2). TGF-β had a predominant role in the heterospheroids as shown by the concomitant inhibition of the TGF-β and the PDGF receptors: the dimension of these heterospheroids was comparable to that obtained with the TGF-βR inhibitor and no additive or synergic effects were visible (Figure 2). Taken together, the transwell and the spheroid models indicate that the NIH-3T3 cells proliferate more when co-cultured with the 4T1 cells than alone, thus showing one of the functional responses typical of activated fibroblasts [30]. This is a paracrine effect of TGF-β and, to a lesser extent, also of PDGF, both of which participate in the reprogramming of resident fibroblasts into CAFs induced by cancer cells [9,31,32].

The role of TGF-β and PDGF as the main paracrine factors promoting fibroblast activation was confirmed in heterospheroids consisting of human TNBC cells and human fibroblasts (MDA-MB-231 and MRC-5 cells, respectively). Such heterospheroids were cultured with either the TGFβR or the PDGFR inhibitor, or both inhibitors, for 7 days and showed a significantly decreased diameter upon exposure to either the TGFβR or PDGFR inhibitor (Appendix A). In this model, PDGF seemed to be more impactful than in the mouse model and to be functionally more prominent than TGFβ, as witnessed by the further decrease in size observed upon exposure to both inhibitors. In any case, the similar responses obtained using the murine and the human models suggest that our approach would be applicable to a wide array of cancer cells-fibroblasts pairs.

#### 3.1.2. Enhanced Release of IL-6 by Activated Fibroblasts

Activated fibroblasts secrete cytokine IL-6 [20]; thus, we measured the concentration of IL-6 in the medium of both transwell co-cultures and heterospheroids to assess whether the 4T1 cells affect the status of the fibroblasts.

The NIH-3T3 cells on transwells responded to TGF-β by significantly increasing IL-6 release with respect to non-treated cells (Figure 3a). As expected, the TGF-βR inhibitor blunted this response (Figure 3a). Similar results were obtained with PDGF, even though CP-673451 produced only a partial inhibition (Figure 3a). Interestingly, SB431542 and CP-673451 had no impact on IL-6, even at a concentration >100-fold higher than the IC_50_ (94 nM and 1 nM, respectively) (Figure 3b). Instead, inhibition of the SMAD3 pathway produced a highly significant drop in the amount of secreted IL-6 (Figure 3b). These observations suggest that multiple signaling pathways, including those activated by TGF-β and PDGF, act redundantly and converge on SMAD3 to control IL-6 secretion.

To further elucidate the role of TGF-β and PDGF on IL-6 release by activated fibroblasts, the same experiment was repeated on the spheroids. We confirmed that NIH-3T3 monospheroids are responsive to TGF-β and PDGF, and to their inhibitors (Figure 4a). The increased IL-6 levels in the medium of the heterospheroid vs. the monospheroid cultures strengthen the notion that the tumor cells are sufficient to elicit this CAF feature in the fibroblasts in our culture settings. At variance with the transwell model, the enhanced release of IL-6 from activated NIH-3T3 cells in the spheroids could be fully reverted by either the TGF-βR, the PDGFR, or the SMAD3 inhibitor (Figure 4b). Differences in the available concentration of the corresponding growth factors and/or their signaling could account for this discrepancy.

Importantly, a similar response was observed in the human spheroids (Appendix A). The human heterospheroids released more IL-6 than the MRC-5 monospheroids and the each of the three inhibitors significantly reduced IL-6 levels, although only a partial reversion could be observed. These results corroborate the notion that our approach may be valuable to study other TNBC models.

The sum of these results shows that TNBC cells are sufficient to activate fibroblasts and for them to acquire key CAF phenotypes. Furthermore, it appears that both TNBC cell-derived TGF-β and PDGF are involved in fibroblasts activation in these settings.

### 3.2. Activated Fibroblasts Induce Aggressive Phenotypes in 4T1 Cells

#### 3.2.1. IL-6-Mediated Chemoresistance

Chemoresistance is determined by both intrinsic factors such as genomic aberrations in cancer cells and extrinsic factors present in the TME [33]. IL-6 is released by various cell types present in the TME, including activated fibroblasts [34] (Figure 3 and Figure 4), and promotes chemoresistance in tumor cells [21,22].

We used our transwell model and doxorubicin (DOX) to determine the level of chemoresistance of 4T1 cells, alone or co-cultured with NIH-3T3 fibroblasts, with or without IL-6. The viability of 4T1 cells treated with either 0.1 or 1 μM DOX for 48 h was only 20-percent of the control culture. Strikingly, the 4T1 cells co-cultured with the fibroblasts became fully resistant to the cytotoxic effects of 0.1 μM DOX and an almost complete protection could be observed at 1 μM DOX (Figure 5). Thus, our model recapitulates the protective effects of activated fibroblasts on breast cancer cells, suggesting that it may be useful to study chemo-resistance mechanisms. We also investigated the role of IL-6 in 4T1 chemoresistance, comparing resistance to DOX in 4T1 monocultures treated with soluble IL-6 to that of the co-cultures. IL-6 significantly increased 4T1 cell viability at the lower DOX dose and had a less prominent effect at the higher one (Figure 5). Hence, IL-6 appears to contribute to the chemoresistance of 4T1 monocultures but is insufficient to confer full protection. The difference between the paracrine effects elicited by IL-6 and by NIH-3T3 cells suggests that multiple cytokines and soluble factors cooperate in mediating CAF-induced chemoresistance in cancer cells.

DOX-induced variations in 4T1 cell number were quantified in the spheroid model, pre-labeling 4T1 cells with Hoechst and then measuring changes in their fluorescence intensity. One μM DOX was selected because no significant reduction in the Hoechst signal was detected at lower concentrations. DOX caused disaggregation of 4T1 monospheroids accompanied by a notable reduction in fluorescence signal (Figure 6). Conversely, DOX had no effect on IL-6-treated monospheroids nor on the heterospheroids. Quantification of the fluorescence of Hoechst-labeled 4T1 nuclei revealed a significant drop (−50%) in the DOX-treated with respect to control spheroids (Figure 6). In keeping with the effects that fibroblasts exerted on the 4T1 cells in the transwells, the heterospheroids treated with DOX did not show any appreciable reduction in fluorescence intensity, and IL-6 conferred full protection on 4T1 monospheroids to DOX. The fact that the fluorescence intensity of Hoechst-labeled 4T1 nuclei was lower in DOX-treated heterospheroids cultured with the TGF-βR inhibitor than in those without this inhibitor further corroborates that activated fibroblasts promote 4T1 chemoresistance (Appendix A).

Together, these data show that activated fibroblasts make a fundamental contribution to making TN mammary cancer cells resistant to DOX and further suggest that IL-6 is a main player.

#### 3.2.2. Tumor Invasiveness

CAF-induced chemoresistance in tumor cells is often associated with tumor progression and metastasis [4]. Notably, IL-6 has been identified as one of the cytokines involved in these events [18]. Therefore, we analyzed the proliferation and the migratory abilities of 4T1 cells co-cultured with either activated NIH-3T3 cells or with soluble IL-6.

The proliferation of 4T1 was evaluated in three different experimental conditions: monoculture (control), monoculture in the presence of IL-6, or co-culture with NIH-3T3 cells on transwell. We found that tumor cells cultured with activated fibroblasts or with IL-6 proliferated more than the control ones (Figure 7a).

The bioluminescence of luciferase+ 4T1 cells was used to verify that activated fibroblasts promote cancer-cell proliferation in heterospheroids as compared with 4T1 monospheroids (Appendix A). The bioluminescence intensity (BLI) of 4T1 cells co-cultured with NIH-3T3 cells was 3-fold higher than that detected in 4T1 monospheroids. This suggests that 7-day-old heterospheroids contain a higher number of tumor cells than the monospheroids, even though the same number of 4T1 cells was employed to generate the spheroids.

Transwell cultures were also used to assess the migration of 4T1 cells by means of wound-healing assays performed after detachment and reseeding on culture-insert Petri dishes. The migration of the 4T1 cells in the free space was monitored (Figure 7b) and the gap between the two edges was measured over time (Figure 7c). The NIH-3T3 cells promoted 4T1 cell migration with respect to control 4T1 cells, from 4 h onwards, whereas IL-6 had no such effect (Figure 7c).

Next, we took advantage of migration assays in which 4T1 cells grown with the NIH-3T3 cells on transwell were harvested and reseeded on a 8 µm-pore insert of a transwell unit. Their migration to the lower side of the insert was determined 24 h later: the results strengthened the notion that activated fibroblasts, but not IL-6, promote 4T1 cell migration (Appendix A). Therefore, IL-6 seems to affect only 4T1 cell proliferation, although some previous studies demonstrated that IL-6 secreted by CAFs can increase both proliferation and migration of cancer cells [35,36,37]. Our results instead agree with other reports suggesting that the proliferation and migration are not necessarily coupled in cancer cells [38]. Nevertheless, other soluble factors could be involved in enhancing the migration of TNBC cells upon interaction with activated fibroblasts [39,40].

## 4. Conclusions

The interplay between resident fibroblasts and tumor cells in solid tumors is a promising target for developing more effective anti-cancer therapies. As such, the identification of the factors regulating this complex process and the subsequent detailed characterization of their mechanisms of action are extremely important. So is also the development of better tools that would allow investigating the interplay between tumor cells and fibroblasts using a reductionist approach and pathologically relevant tractable systems.

With this objective in mind, we developed and validated the use of advanced cell culture models as tools to study the interaction between tumor cells and resident fibroblasts, taking mouse mammary triple-negative tumor cells and mouse fibroblasts as a case study. We employed two different advanced co-culture systems, one permitting only paracrine signaling (transwell model), the other both paracrine and cell-contact-based signaling (spheroid model). Taking advantage of these tools, we unmasked how TGF-β, PDGF and IL-6 mediate the interplay between mammary tumor cells and fibroblasts. We found that the fibroblasts were activated by the TGF-β and PDGF secreted by the tumor cells. This resulted in fibroblasts showing enhanced proliferation and IL-6 secretion. In turn, IL-6 secreted by activated fibroblasts boosted tumor-cell proliferation and chemoresistance, but not migration. Similar conclusions could be reached using human TNBC cells and human fibroblasts.

In conclusion, TNBC cells and fibroblasts make up breast cancer avatars that possess a high level of complexity, which mimics that observed in vivo. Therefore, our advanced co-cultures provide a pathologically relevant tractable system to study the role of the TME in breast cancer progression with a reductionist approach. The fact that they do not require either ECM or solid scaffolds makes them simpler and cheaper than other alternative methods [41,42,43]. Of course, such an approach does not allow for the implantation of the resulting 3D cultures or direct monitoring of cancer cells leaving the spheroids.

While the exact nature of activated fibroblasts would need to be characterized further using in situ single-cell analyses, we anticipate that our advanced cell culture models will become increasingly utilized to dissect the interplay between tumor cells and the TME both in basic and preclinical research in the years to come.

## Figures and Tables

**Figure 1 cancers-15-02498-f001:**
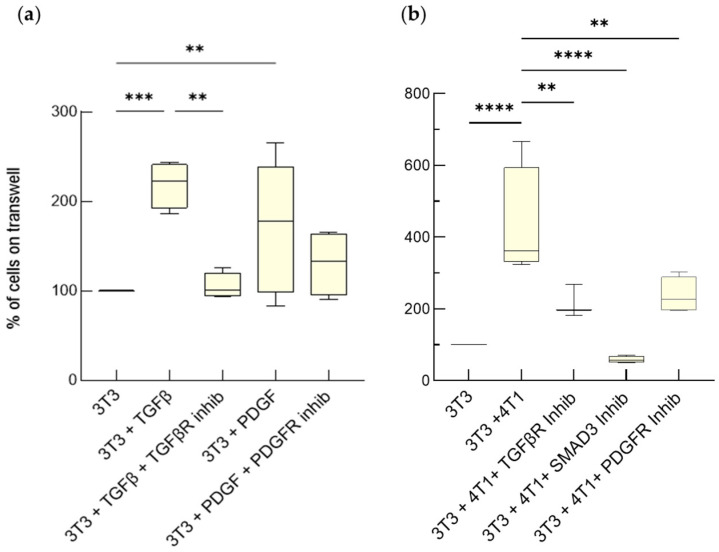
TGF-β and PDGF released by 4T1 cells promote fibroblast proliferation. NIH-3T3 fibroblasts (3T3) on the upper side of transwell upon exposure to TGF-β (5 ng/mL), in the absence or presence of 10 μM TGF-β receptor inhibitor (SB431542) or to PDGF (50 ng/mL) in the absence or presence of 1 μM of the PDGFRα/β inhibitor (CP-673451) (**a**), or to 4T1 cells in the absence or presence of the same inhibitors and of SMAD3 inhibitor (SIS3) (**b**) were counted at day 5 of culture. The number of untreated NIH-3T3 cells in monoculture was set as reference at 100%. Data, represented as box-and-whisker plots, were compared by One-way ANOVA (**** *p* < 0.0005, *** *p* < 0.001, ** *p* < 0.01; *n* = 4–14).

**Figure 2 cancers-15-02498-f002:**
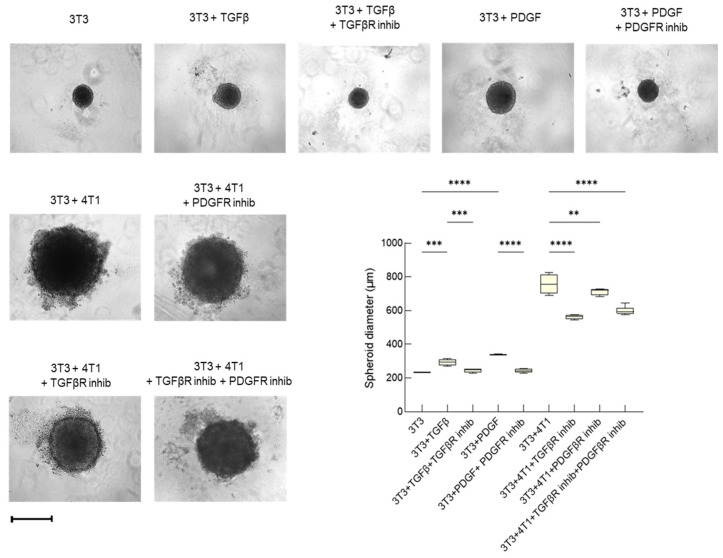
TGF-β and PDGF promote fibroblast proliferation in mono- and hetero-spheroid cultures. Diameter of NIH-3T3 cell (3T3) monospheroid cultures treated with TGF-β (5 ng/m)L for 7 days, in the absence or presence of 10 μM TGF-βR inhibitor (SB431542), or with PDGF (50 ng/mL), in the absence or presence of 1 μM PDGFRα/β inhibitor (CP-673451), was measured at day 7; diameter of 7-day heterospheroids (3T3/4T1) in the absence or presence of the same inhibitors. Diameter of the spheroids was determined using the acquired images. Data, represented as box-and-whisker plots, were compared by One-way ANOVA (**** *p* < 0.0005, *** *p* < 0.001, ** *p* < 0.01; *n* = 4). Bar: 500 µm.

**Figure 3 cancers-15-02498-f003:**
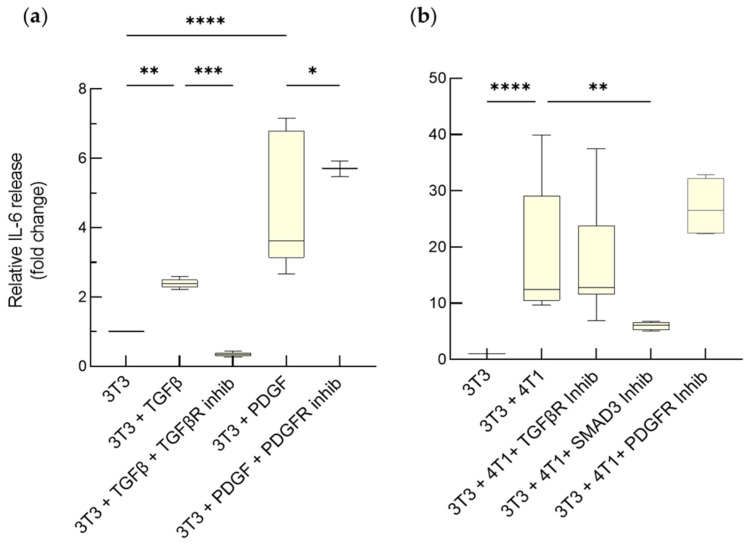
TGF-β and PDGF promote IL-6 release by activated fibroblasts on transwell. Variation of IL-6 release from NIH-3T3 fibroblasts, cultured on the upper side of transwell, at the end of 5 days exposure to TGF-β (5 ng/mL), in the absence or presence of 10 μM TGF-β receptor inhibitor (SB431542), or to PDGF (50 ng/mL) in the absence or presence of 1 μM of the PDGFRα/β inhibitor (CP-673451) (**a**), or to 4T1 cells in the absence or presence of the same inhibitors and of SMAD3 inhibitor (SIS3) (**b**). Untreated 3T3 cells in monoculture have been considered as reference (1). Data, represented as box-and-whisker plots, were compared by One-way ANOVA (**** *p* < 0.0005, *** *p* < 0.001, ** *p* < 0.01, * *p* < 0.05; *n* = 4–14).

**Figure 4 cancers-15-02498-f004:**
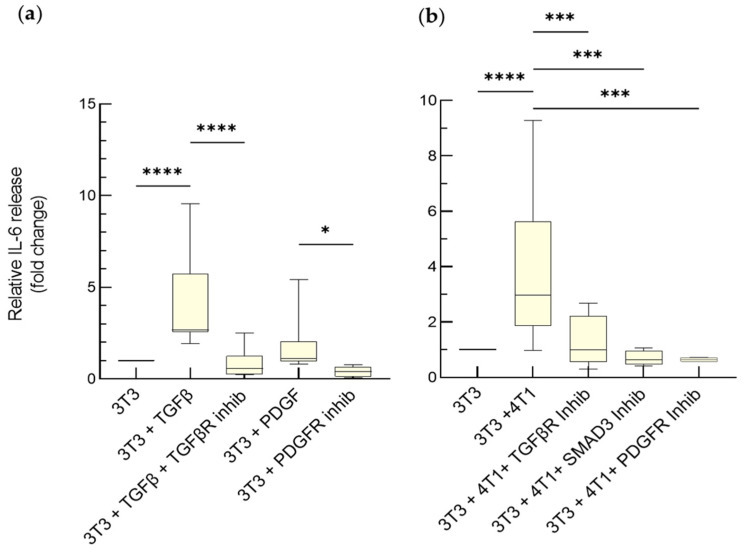
TGF-β and PDGF promote IL-6 release by activated fibroblasts in mono- and hetero-spheroids. Variation of IL-6 release from monospheroids (3T3) exposed to TGF-β (5 ng/mL) in the absence or presence of 10 μM TGF-β receptor inhibitor (SB431542), to PDGF (50 ng/mL) in the absence or presence of 1 μM PDGFRα/β inhibitor (CP-673451) (**a**), or to 4T1 cells in the absence or presence of the same inhibitors and of SMAD3 inhibitor (SIS3) (**b**). Untreated 3T3 monospheroids were considered as the reference (1). Data, represented as box-and-whisker plots, were compared by One-way ANOVA (**** *p* < 0.0005, *** *p* < 0.001, * *p* < 0.05; *n* = 4–15).

**Figure 5 cancers-15-02498-f005:**
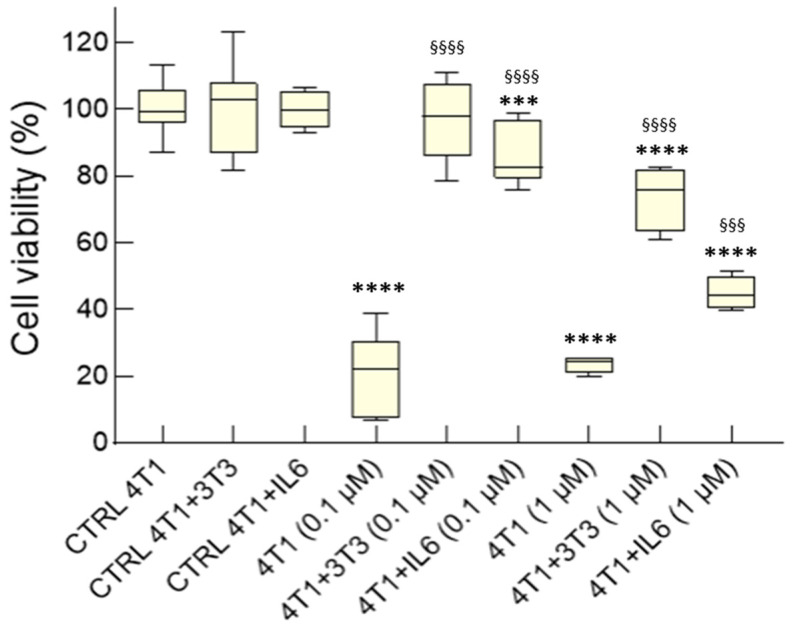
Chemoresistance of 4T1 cells exposed to activated fibroblasts on transwell and role of IL-6. Cell viability of 4T1 cultured for 5 days with or without NIH-3T3 fibroblasts in transwell, or with or without IL-6 (50 ng/mL), and then treated with 0.1 or 1 μM DOX for 48 h. The viability of untreated 4T1 monocultures was set as 100%. Data, represented as box-and-whisker plots, were compared by One-way ANOVA (**** *p* < 0.0005, *** *p* < 0.001, vs. CTRL; ^§§§§^
*p* < 0.0005, ^§§§^
*p* < 0.001 vs. 4T1; *n* = 4–18).

**Figure 6 cancers-15-02498-f006:**
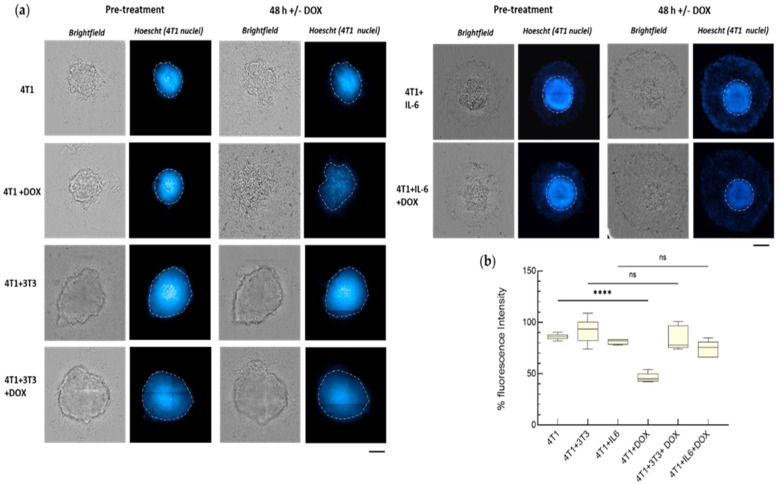
Chemoresistance of 4T1 cells in heterospheroids and role of IL-6. (**a**) Operetta CLS images of 4T1 monospheroids cultured for 7 days in the absence or presence of IL-6 (50 ng/mL), and of 7-day-old heterospheroids (4T1/3T3), before and after 48 h incubation with 1 μM DOX. 4T1 cells were labeled with the nuclear dye Hoechst 33342 (10 μM) before spheroid formation. (**b**) Percentage fluorescence intensity of 4T1 nuclei in the spheroids at 48 h of DOX treatment, compared to the control samples. FI before the treatment was taken as a reference (100%). FI was determined for the ROI delimited by white dotted lines. Data, represented as box-and-whisker plots, were compared by One-way ANOVA (**** *p* < 0.0005 vs. untreated spheroids; ns: not significant; *n* = 5–6). Bar: 200 µm.

**Figure 7 cancers-15-02498-f007:**
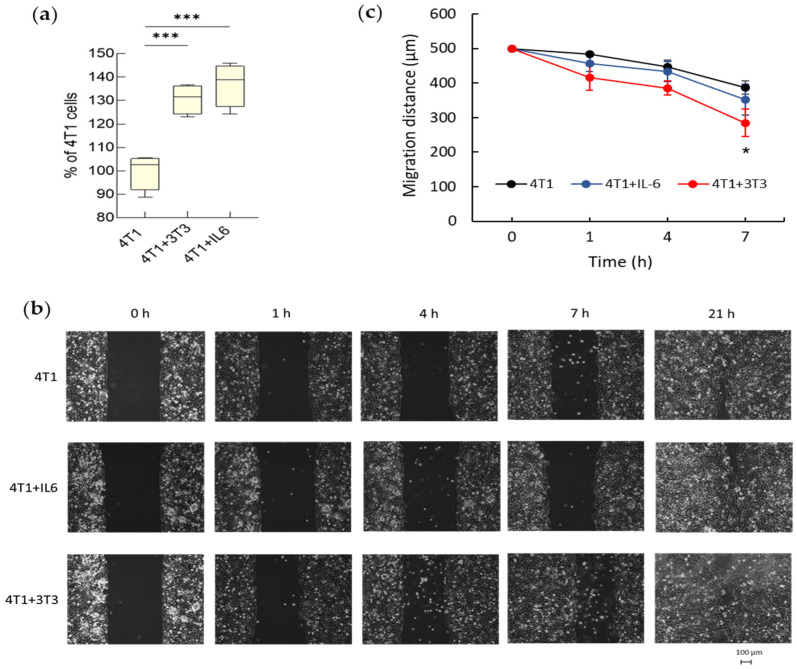
Proliferation and migration of 4T1 cells exposed to activated fibroblasts on transwell and role of IL-6. (**a**) Percentage of 4T1 cells counted in a single well of 12-well multiplate at the end of 5 days of culture in the presence of 50 ng/mL IL-6, or co-culture with 3T3. Untreated 4T1 cells in monoculture have been considered as 100%. Data, represented as box-and-whisker plots, are compared by One-way ANOVA (*** *p* < 0.001 vs. control 4T1, n = 4) (**b**) Wound-healing assay performed to detect the migration of cells at 1, 4, 7 and 21 h; starting cell-free space: 500 μm. (**c**) Migration-free space (μm) over time (0–7 h). Points represent Mean values ± SE (*n* = 3) and were compared by One-way ANOVA (* *p* < 0.05 vs. control 4T1).

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
