# Peer review of "Advanced Cell Culture Models Illuminate the Interplay between Mammary Tumor Cells and Activated Fibroblasts"

_cancers, 2023, doi:10.3390/cancers15092498_

Round 1

Reviewer 1 Report

The Paragraphs Are Too Many In the Introduction, Kindly Reduce

The Methodology Should Be Extensively Explained.

Identify The Limitation And Future Directions.

The authors proposed an advanced cell culture models to illuminate the interplay between mammary tumor cells and activated fibroblasts which are known to be highly aggressive and difficult to eradicate. Breast carcinomas have drawn lots of attention recently to the academic field as it is referred to as the second most deadly disease among women. This topic is of an interest to the research community thus the authors are advised to address the below concerns to improve the quality of their manuscript.

1.) The flowchart of the proposed model should be presented for easy understanding.

2.) The architecture of the model should be drawn to illustrate more on the contribution of the manuscript.

3.) The last two Paragraphs of the introduction should include a bullet point of the contribution of the manuscript as well as the structure of the paper.

4.) Authors are advised to compare their results with existing methodologies.

Reviewer 2 Report

The article aimed at using advanced cell culture models to study the interplay between mammary tumor cells and activated fibroblasts. The authors found the IL-6 secreted by activated fibroblasts enhanced tumor cell proliferation and chemoresistance. The authors pointed out that advanced co-culture models provide a pathologically relevant tractable system to study the role of the TME in breast cancer progression with a reductionist approach. The authors employed two different co-culture systems, transwell model and spheroid model. 

In general, the manuscript was well-written. The language is clear and professional. The experiments were well-designed.

The author used the murine triple-negative breast cancer 4T1 cell line and murine embryonic fibroblasts NIH-3T3 cell line to build the co-culture system. It would be better also to include human cell lines or patient-derived cell lines. Also, in figure 1, the authors would like to show the TGF-b and PDGF released by 4T1 cells promote fibroblast proliferation. It would be better to include conditions "3T3+PDGF+PDGFR inhibitor" and "3T3+SMAD3 inhibitor" in figure 1a for completeness. Similar improvements can also be adapted in Fig. 3 and 4.

In addition, there are some small places can be improved in the manuscript.

Line 16 TGF-b is not correctly formatted.

Line 185 "4T1 cell" is not correctly formatted. Similar problem in line 245.

The terminology "NIH-3T3" is not consistently written in the manuscript. The author used "NIH3T3" and "NIH-3T3" interchangeably. 

The format of Figure 5 and 6a is quite different from figures 1 to 4. It would be better to be the same format.

Reviewer 3 Report

The reviewer cannot understand the novelty. Many papers have been on 3D models based on the interaction between cancer cells and fibroblasts. In addition, many technologies would be introduced to realize the interaction close to in vivo. The authors must introduce the recent research and discuss the novelty by quoting these references. In this version, the reviewer cannot recommend the publication. The manuscript would be re-considered for publication only when all the comments were responded.

1.

There are many papers on CAFs and cancer models. In the current version, it is difficult to understand the strength of this study. The authors must mention the novelty by comparing these preferences (not limited to breast cancer ).

Review (for concept)

https://doi.org/10.1002/adhm.202000608

Cancers 202012(10), 2754

Research papers

Chip  https://doi.org/10.1016/j.biomaterials.2015.03.012

DDS  https://doi.org/10.1016/j.reth.2020.02.003

Scafflods https://doi.org/10.1016/j.biomaterials.2017.12.017

2. Figure 2

The size of spheroids is too large. So why is the system able to handle it? Why was it survive?

3.

The authors should investigate the gene or protein expression related to breast cancers. Only viability, but the characteristics of spheroids cannot be understood.

Author Response

-The authors must introduce the recent research and discuss the novelty by quoting these references. In this version, the reviewer cannot recommend the publication. Suggested articles: https://doi.org/10.1002/adhm.202000608;  https://doi.org/10.1016/j.biomaterials.2015.03.012 , https://doi.org/10.1016/j.reth.2020.02.003, https://doi.org/10.1016/j.biomaterials.2017.12.017

Our approach does not require either gels or scaffolds, therefore being simpler and cheaper than the other two. Of course, such a reductionistic approach does not allow for implantation of 3D entities or tracking cancer cells leaving the spheroids. It is well acknowledged in the field that all these tumor avatars have limitations. As such, systems should be chosen according to the specific question that should be addressed. This issue has been faced in Conclusions and the review suggested by the reviewer cited in this context (ref…)

  1. Figure 2. The size of spheroids is too large. So why is the system able to handle it? Why was it survive?

We find it important to highlight that the spheroids were cultured in medium and not in Matrigel, thus favoring diffusion of gas and nutrients. However, the reviewer’s point is well taken. Hence, we have  added a Supplementary Figure (S 1b) showing the morphology of nuclei labelled with Hoechst in a central section taken from a Z-stack of a representative heterospheroid. The images indicate that no hollow core or dead cells are visible. This is commented in the revised text (lines 294-296).

3.

The authors should investigate the gene or protein expression related to breast cancers. Only viability, but the characteristics of spheroids cannot be understood.

We respectfully feel that analyzing gene or protein expression is beyond the scope of this study, let alone that this would require in situ single-cell analyses, the setup of which would be itself a project.

Reviewer 4 Report

This manuscript shows the interaction between 4T1 cell lines and activated NIH3T3 cell lines. The authors performed the co-culture of these cell lines with inhibitor treatment by They assessed the cancerous phenotypes and gene expression. Their results showed a part of molecular mechanism in co-culture between cancer cells and CAF-like cells. There are several concerns for the publication.

Comments

Major

1. There are many similar previous studies. The measurement system may be technically novel method, however, the discovery itself has poor novelty. The authors should be more systematic and rigorous in comparing their observations with what is currently known.

2. The authors used "Pro-Tumorigenic phenotype" in the manuscript. What is "Pro-Tumorigenic"? It's co-cultured with cancer cells, so it's probably not "Pro-". In addition, the authors should evaluate the function of activated NIH3T3 cells, following previously reported CAF Classifications and activities.

3. In the experiments of inhibitor treatment in spheres, the authors should evaluate the consists of sphere in more detail. Do the inhibitor treatments affect the consist of sphere?

4. The authors performed wound healing assay with 2D culture. In general sphere culture showed cancer cell migration from sphere. In this manuscript, the reviewer wonders that cancer cell migration in the sphere culture is not observed. The authors should evaluate the migration activities with 3D culture system.

5. The CAF model using NIH-3T3 cells has utilized previously, however, there are many lacks as a CAFs model. Their study design has a lot of limitations. The authors should clarify all limitations of the presented models and cell lines. In addition, they performed the experiments using only one cancer cell line. The authors should confirm the phenotypes using other cancer cell lines.

Minor

6. The manuscript needs the English proofing.

Author Response

  1. There are many similar previous studies. The measurement system may be technically novel method, however, the discovery itself has poor novelty. The authors should be more systematic and rigorous in comparing their observations with what is currently known.

The goal of this project was to build and validate advanced cell culture models as tools to study the interplay between mammary tumor cells and fibroblasts from the tumor microenvironment.

We took mouse mammary triple-negative tumor cells and mouse fibroblasts only as a case study, which has allowed us to decipher the involvement of TGF-β, PDGF and IL-6 in the interplay between tumor cells and fibroblasts. In the revised version, we have extended the breath of our conclusions to human models consisting of human TNBC cells and human fibroblasts (Figure S2 and S3).

We posit that advanced co-cultures provide a pathologically relevant tractable system to study role of the TME in in cancer progression by a reductionist approach. We agree with the reviewer that the main advance brought about by our work is a technological one and firmly believe that it will find wide application in the field.

  1. The authors used "Pro-Tumorigenic phenotype" in the manuscript. What is "Pro-Tumorigenic"? It's co-cultured with cancer cells, so it's probably not "Pro-". In addition, the authors should evaluate the function of activated NIH3T3 cells, following previously reported CAF Classifications and activities.

The adjective “pro-tumorigenic” is often coupled with the substantive “phenotype” in the literature about CAFs and is meant to indicate properties that allow CAFs to promote tumor onset and progression. Indeed, we have previously reported key CAF activities, namely the ability to enhance proliferation (Figure 7a), invasiveness (Figures 7b and c), and chemoresistance (Figures 5 and 6) of the tumor cells they associate with. New experiments were performed to strengthen this point on revision (see Fig S6).

In situ single cell analyses, which are beyond the scope of this project, would be required to substantiate this claim. Therefore, we refrained from making speculations or adding this information in the text.

  1. In the experiments of inhibitor treatment in spheres, the authors should evaluate the consists of sphere in more detail. Do the inhibitor treatments affect the consist of sphere?

The diameter of the spheroids is reduced in the presence of the inhibitors, even if cancer cell migration out of the sphere is still visible (Figure 2). Changes in spheroid consistence were not obvious.

  1. The authors performed wound healing assay with 2D culture. In general sphere culture showed cancer cell migration from sphere. In this manuscript, the reviewer wonders that cancer cell migration in the sphere culture is not observed. The authors should evaluate the migration activities with 3D culture system.

Cell migration from sphere is visible in the Paula images shown in Figure 2, although those settings did not allow assessing which cell type left the spheroid. More importantly, spheroids were not embedded in Matrigel or any other ECM, precluding a faithful characterization of cell migration in the 3D cultures.

  1. The CAF model using NIH-3T3 cells has utilized previously, however, there are many lacks as a CAFs model. Their study design has a lot of limitations. The authors should clarify all limitations of the presented models and cell lines. In addition, they performed the experiments using only one cancer cell line. The authors should confirm the phenotypes using other cancer cell lines.

We showed that the fibroblasts acquire properties of genuine CAFs under the experimental conditions employed in our study. On revision, we have validated a few key findings using a human TNBC-fibroblast pair. Our system, just like all in vitro approaches, has limitations and have added a disclaimer in the conclusions. Nevertheless, we trust that our advanced cell culture models will benefit the field.

Minor Remarks

English has carefully been proofed.

Round 2

Reviewer 3 Report

I recommend the publication.

Author Response

We thank the reviewer for supporting the publication of our study

Reviewer 4 Report

The manuscript is improved, however, that is still riddled with some serious issue.

Comments

1. The revised supplementary materials lacks Figure S2 and S3 that shows the experiments using human cell lines. The reviewer cannot judge the results, therefore, it is should be decided as revision. The reviewer thinks that it is necessary to judge again after confirmation of all Figures.

2. In previous comment 1, 3, 4, and 5, the reviewer thinks that the authors have not performed proper experiments.

3. The reviewer can not understand the novelty and advantage of method. The previous method seems to be superior. It should be clarified exactly what is the merit of this method. The reviewer is not satisfied with their explanations and results. The manuscript keeps the potential to reject.
